# Protein nanowires with tunable functionality and programmable self-assembly using sequence-controlled synthesis

Daniel Mark Shapiro [1,2,3,4], Gunasheil Mandava[3,4], Sibel Ebru Yalcin [3,4], Pol Arranz-Gibert[1,2], Peter J. Dahl[3,4], Catharine Shipps [3,4], Yangqi Gu [3,4], Vishok Srikanth [3,4], Aldo I. Salazar-Morales [3,4], J. Patrick O'Brien[3,4], Koen Vanderschuren[1,2], Dennis Vu [3,4], Victor S. Batista [5], Nikhil S. Malvankar [3,4,7 ✉] & Farren J. Isaacs [1,2,6,7 ✉]

Advances in synthetic biology permit the genetic encoding of synthetic chemistries at monomeric precision, enabling the synthesis of programmable proteins with tunable properties. Bacterial pili serve as an attractive biomaterial for the development of engineered protein materials due to their ability to self-assemble into mechanically robust filaments. However, most biomaterials lack electronic functionality and atomic structures of putative conductive proteins are not known. Here, we engineer high electronic conductivity in pili produced by a genomically-recoded *E. coli* strain. Incorporation of tryptophan into pili increased conductivity of individual filaments >80-fold. Computationally-guided ordering of the pili into nanostructures increased conductivity 5-fold compared to unordered pili networks. Site-specific conjugation of pili with gold nanoparticles, facilitated by incorporating the nonstandard amino acid propargyloxy-phenylalanine, increased filament conductivity ~170-fold. This work demonstrates the sequence-defined production of highly-conductive protein nanowires and hybrid organic-inorganic biomaterials with genetically-programmable electronic functionalities not accessible in nature or through chemical-based synthesis.

[1] Department of Molecular, Cellular & Developmental Biology, Yale University, New Haven, CT 06520, USA. [2] Systems Biology Institute, Yale University, West Haven, CT 06516, USA. [3] Department of Molecular Biophysics and Biochemistry, Yale University, New Haven, CT 06520, USA. [4] Microbial Sciences Institute, Yale University, West Haven, CT 06516, USA. [5] Department of Chemistry, Yale University, New Haven, CT 06520, USA. [6] Department of Biomedical Engineering, Yale University, New Haven, CT 06520, USA. [7] These authors contributed equally: Nikhil Malvankar, Farren J. Isaacs. ✉email: nikhil.malvankar@yale.edu; farren.isaacs@yale.edu

Materials produced from synthetic chemical processes provide access to a broad range of chemical structures yet are constrained by the lack of sequence-defined polymerization methods. In contrast, biological systems employ sequence-controlled processes to synthesize biomolecules, in which the molecular information encoded by nucleic acids is converted into sequence-controlled protein polymers[1]. Protein polymers have evolved to assume specialized functions in nature, among which is the formation of dynamic protein-based materials (e.g., collagen, silk, and elastin). These multifunctional materials possess versatile functions spanning a range of strength, elasticity, and stability, but lack electronic or optical functionality. Engineered living materials with programmable functionalities and environmental resilience are attractive biomaterials due to their ability to regenerate, sense, and adapt to environmental cues[2]. However, nature is constrained to a small set of organic monomeric building blocks, the 20 canonical amino acids, thereby limiting the chemical diversity of polymeric biomaterials. Expanding the chemical palette of genetically encoded chemistries could yield new classes of enzymes, materials, and therapeutics produced in a sequence-defined manner with diverse chemistries.

Many bacteria produce filamentous protein appendages on their surface called pili, which are critical to bacterial infections due to their roles in host colonization and surface sensing[3], bacterial motility[4], and natural competence[5]. In addition to their biomedical importance, pili filaments are attractive biomaterials due to their capacity to self-assemble through natural polymerization while retaining extraordinary mechanical stability and robustness, being able to withstand a wide range of temperatures, pH, and protein-denaturing agents such as SDS and urea[6–9]. However, there are several major challenges in the use of pili as multifunctional biomaterials. First, like most proteins, pili lack electronic or optical functionality, which are critical for the development of next-generation bioelectronics. It was previously thought that some soil bacteria such as *Geobacter sulfurreducens* produce conductive type IV pili. However, structural studies revealed that conductive filaments on the bacterial surface are polymerized cytochromes whereas pili remain inside the cell and are involved in the secretion of filamentous cytochromes[10–14]. *G. sulfurreducens* pili are bipartite comprised of the PilA-N and PilA-C proteins, show overall structure similar to the type IV pili[15], but extend past the bacterial surface only when artificially overexpressed, and show very low conductivity[10]. A previous study has claimed conductivity in synthetic pili[16], however the conductivity of the individual synthetic filaments has not been demonstrated along their length, only across their diameter, providing no evidence that the pili could be conductive down their length like a nanowire. Furthermore, their exact biochemical composition is unknown as discussed at length in a previous study[10]. Second, structures of these putative conductive "pili are not available, hindering the elucidation and prediction of structure-function correlations. The final obstacle in using pili as multifunctional biomaterials is that, although in vitro assembly of conductive proteins is feasible, they tend to aggregate, which is not suitable for mass production[16]. Therefore, novel methods for in vivo biomaterial production are required for large-scale production of pili biomaterials endowed with tunable electronic and mechanical functionalities.

In this study we address these challenges by pursuing three strategies that demonstrate large-scale production of conductive pili proteins with tunable electronic properties. First, we strategically encode standard aromatic amino acid mutations into *E. coli* type 1 pili (Fig. 1a, b) using a cryo-EM structure of *E. coli* type 1 pili as a guide (PDB: 6c53)[9,17], and demonstrate mutation-dependent 10- to 84-fold increases in the conductivity of the filaments. Importantly, our work utilizes methods to measure the conductivity of individual filaments rather than filament films, which is distinct to prior conductivity studies of protein networks where high contact resistance between filaments and measurement electrodes has masked the intrinsic electronic properties of the individual filaments[18–20]. Our approach also eliminates network artifacts where conductivity is dominated by percolation behavior, and reduces the impact of inter-filament contact resistance on the conductivity measurements[21]. Second, we engineer long-range conductivity over the micrometer scale by generating networks of hierarchical assemblies of conductive pili using molecular recognition self-assembly[22] (Fig. 1c). Finally, we use a genomically recoded strain of *E. coli*[23] to genetically encode the nonstandard amino acid (nsAA) propargyloxy-phenylalanine (PrOF) to generate pili that form a click-chemistry-functional scaffold for the precise, site-specific conjugation of gold nanoparticles (AuNPs), leading to the biosynthesis of sequence-controlled organic-inorganic hybrid biomaterials endowed with conductivity enhanced by 170-fold (Fig. 1d).

## Results

**Engineered biomaterial platform**. Type 1 pili are micrometer-long helical polymers comprised of thousands of copies of the FimA protein[3,7,9,17,24] and can be purified in large quantities at high purity (Fig. 1a, right, Supplementary Fig. 1). We used *E. coli* as a chassis organism because it is easy to grow, easy to genetically manipulate, can incorporate nsAAs, and expresses type 1 pili naturally (Fig. 1a, left). The *E. coli* strain we employed to express all type 1 pili variants was a previously described *E. coli* MG1655 derivative that enables accurate insertion of nsAAs into proteins with high efficiency[23,25,26]. This *E. coli* strain, known as the genomically recoded organism (GRO), has all instances of UAG stop codons re-coded to synonymous UAA codons and lacks release factor 1 (RF1), establishing the UAG codon as an open codon. The open UAG codon, together with an engineered orthogonal translation system (OTS) encoding an orthogonal *M jannaschii* tRNA-aminoacyl-tRNA synthetase pair, allows for the efficient and site-specific incorporation of diverse nsAAs[23,25]. Taken together, this recoded strain of *E. coli* permits the production of proteins with multiple instances of nsAAs at high yield and accuracy[25], setting the stage for functionalizing pili biomaterials with synthetic chemical modalities.

**Aromatic amino acid mutations in type 1 pili increase conductivity on the nanometer scale**. We first sought to engineer *E. coli* pili with enhanced conductivity by generating a channel of aromatic amino acids along the filament. This effort was guided by recent studies which have shown that the close stacking of aromatic tyrosine residues enables electron transfer in individual protein crystals over micrometer-long distances[21]. These studies contrast the widely held belief that proteins are considered to be electronic insulators that can only transfer electrons over a few nanometers[27]. Indeed, several non-aromatic proteins have recently been shown to be conductive over nanometers in single-molecule measurements, with little decay due to distance, provided charge is precisely injected into the protein interior with good contact[28,29]. As type 1 pili are common virulence factors in some pathogenic strains of *E. coli*, it has been observed that many residues on the outside of type 1 pili are highly variable to avoid immune response[24]. Thus, we hypothesized that select external residues would permit mutations without altering their structure substantially. We identified candidate residues that were both in the set of variable surface residues and which, when mutated, would be in close enough proximity (<15 Å) to each other to

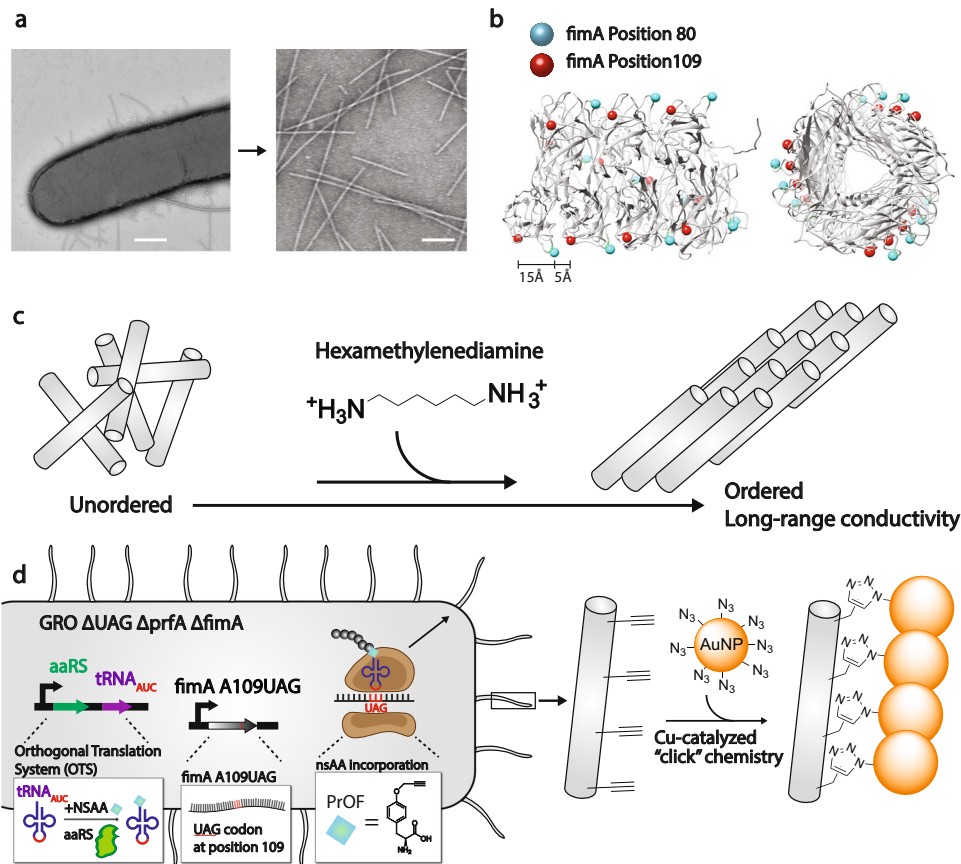

**Fig. 1 Strategy to engineer electronic conductivity into *E. coli* pili nanofilaments. a** Representative TEM image of an *E. coli* cell (left) expressing pili and purified pili (right). Scale bars, 200 nm (left) and 100 nm (right). **b** Cryo-EM structure of 8 FimA monomers forming mature pilus. Positions 80 and 109 in each monomer are highlighted with cyan and red spheres respectively. side view (left), front view (right). **a, b** experiments replicated independently greater than 15 times with similar results. **c** Strategy to develop hierarchical ordered structures with enhanced conductivity. **d** Schematic of creating organic-inorganic hybrid pili using gold nanoparticles clicked on through azido-alkyne click chemistry functionality encoded with nsAAs.

facilitate efficient electron transfer, likely through electron hopping, along the pilus: FimA A80, H82, and A109 (Fig. 1b).

We initially performed two assays to confirm the identity of the purified pili encoded by FimA. First, we confirmed that the only purified protein in our samples was FimA using SDS-PAGE gel electrophoresis (Supplementary Fig. 1b). Second, we used an established yeast agglutination assay in which *Saccharomyces cerevisiae* cells agglutinate in the presence of *E. coli* expressing type 1 pili but do not when mixed with *E. coli* that do not express type 1 pili[24,30] (Supplementary Fig. 6). We then used multiplex automated genome engineering[31,32] (MAGE, see methods) to generate a library of FimA mutants in which the native amino acids A80, H82, or A109 were replaced with the aromatic amino acids phenylalanine, tyrosine, histidine, and tryptophan (Supplementary Table 1) and assessed the impact of these mutations on FimA pili expression using Transmission Electron Microscopy (TEM). We found eight mutations in FimA that preserved pili expression and assembly and measured the conductivity of six mutants: A80F, A109F, A80F A109F (double mutant), A109Y, A80Y A109Y (double mutant), and A80W A109W (double mutant) (Fig. 1a, Supplementary Fig. 1, Supplementary Table 1). We made four pili variants with mutants at position 82, and all resulted in the loss of pili expression (Supplementary Table 1). We hypothesized that mutant pili with two additional aromatic residues per monomer would be more conductive than those with only one additional aromatic amino acid per monomer, as the aromatic groups would be in closer proximity to one another and facilitate enhanced electron transport along the pilus.

To determine the conductivity of individual pili filaments, we placed pili on gold electrodes separated by 300 nm nonconductive gaps and located individual filaments bridging two electrodes using atomic force microscopy (AFM, Fig. 2a, b, Supplementary Fig. 2). We then applied voltages ranging from −0.15 V to +0.15 V across the filament and measured the steady-state current through the pili. After measuring the conductance as the slope of the current-voltage curve (Supplementary Fig. 3), we calculated the conductivity as reported previously[33]. As expected, wild-type pili filaments showed very low conductivity (Fig. 2c–e) of $0.51 \pm 0.14$ mS/cm. Incorporating a single phenylalanine mutation (A80F or A109F) in each FimA monomer increased the conductivity of individual pili filaments by 10-fold to $5.168 \pm 0.154$ mS/cm for A80F and $5.933 \pm 0.323$ mS/cm for A109F. Incorporating a single tyrosine mutation (A109Y) in each FimA monomer increased the conductivity of individual pili filaments 18-fold to $9.208 \pm 0.377$ mS/cm (Fig. 2c, e). Incorporating two phenylalanine residues per monomer (A80F A109F) generated pili that were slightly more conductive than the single phenylalanine mutants at $7.250 \pm 0.782$ mS/cm, but still less conductive than pili incorporating tyrosine. Pili incorporating two tyrosine residues per monomer (A80Y A109Y) were only slightly more conductive than those with one tyrosine per monomer at $9.287 \pm 0.504$ mS/cm.

The insights from the phenylalanine and tyrosine mutants, namely that the conductivity with two incorporated residues was higher than the conductivity with one, guided us to investigate pili with two tryptophan residues introduced per FimA monomer

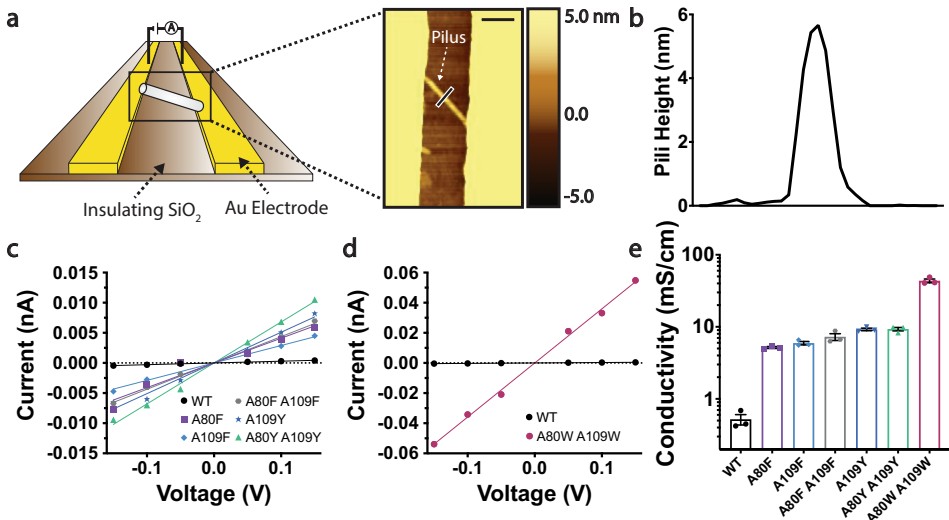

**Fig. 2 Electronic conductivity of individual pili shows 84-fold increase due to tryptophan. a** Schematic of measurements and AFM image of pili bridging the gold electrodes. Scale bar, 200 nm. **b** Height profile of pili at location (black bar crossing pilus) shown in **a**. **c**, **d** Current–voltage profile of pili with different aromatic residue mutations, each line representative of conductivity measurements on one pilus. Representative points were shifted by a constant value such that the slope of the current–voltage curve retained the same value but intercepted at zero for comparison purposes. Raw current–voltage data is provided in Supplementary Fig. 3 and the source file. Currents were measured after applying voltages from −0.15 to 0.15 V in intervals of 0.05. **e** Conductivity comparison of pili. Error bars represent s.e.m. ($n = 3$).

to study the associated conductivity of this engineered pili variant. Remarkably, incorporating two tryptophan mutations per FimA monomer (Fim A A80W A109W) increased conductivity 84-fold to 43.48 ± 4.53 mS/cm (Fig. 2d, e) from wild-type. These results demonstrate an increase in the conductivity of individual pili filaments in the absence of metal co-factors, close amino acid packing, or high potentials typically required to achieve such conductivity in proteins[18,19,34,35]. To our knowledge, this is the highest reported conductivity of a single protein-based filament without metal cofactors based on a protein with known atomic structure. Notably, the measured conductivity is comparable to the reported conductivity of metal-containing *G. sulfurreducens* filaments, comprised of cytochrome OmcS (51 ± 11 mS/cm[11,36]). Such surprisingly high conductivity is likely due to highly π-stacked molecular structures owing to the indole side chain of tryptophan. In contrast to tyrosine, which requires protein environments that favor proton transfer, tryptophan residues can relay electrons at biologically relevant potentials even in protein environments that disfavor proton transfer[21,37,38]. Fluorescence spectroscopy confirmed that tryptophan substitution did not cause significant structural change in pili as the mutated tryptophan remained solvent-exposed in a manner similar to alanine residues in native pili structure (Supplementary Fig. 4). Therefore, our results demonstrate that the incorporation of two tryptophan residues per FimA monomer can increase the conductivity of individual pili filaments up to 84-fold in the absence of metal cofactors, close amino acid packing, or high potentials typically required to achieve such conductivity in proteins[18,19,34,35].

**Computationally-guided design of hierarchical nanostructures at the micrometer scale.** We next sought to build on the development of the mutant filaments with increased conductivity at the nanometer scale by constructing pili-based nanostructures that are conductive at the micrometer scale (Fig. 1c). This effort is inspired by living systems, which create functional materials through hierarchical self-assembly of nanoscale molecules. Although synthetic molecules can be assembled into artificial

nanostructures, bridging from the nanoscale to the macroscale to create functional macroscopic materials remains a challenge[39]. Our strategy for creating filament nanostructures aimed to use the molecule hexamethylenediamine (HMD) to align conductive pili into bundled lattices through molecular recognition-based self-assembly[22] (Fig. 3a). Previous work[22] demonstrated the ability to create different nanostructures by modulating the identity and concentration of inducer. In this study, we chose one inducer, HMD, which was used to create 2D bundled lattices, to investigate the effect of self-assembly on pili material conductivity. HMD is positively charged at both ends and thus is able to promote alignment of negatively-charged pili (Fig. 3a) into larger structures. To evaluate the use of HMD in the formation of ordered nanowire assemblies, we performed molecular dynamics simulations on two monomers of the FimA A80W A109W pili with and without HMD. We found that over the course of a 100 ns simulation orientations of the pilin monomers that were aligned in close proximity with one another were only realized in the presence of 250 mM HMD (Fig. 3b–d), suggesting that HMD could promote association of the filaments. As the MD simulations at 250 mM HMD set the number of HMD molecules to 55 (see: methods), two pilin monomers would set the protein concentration to be approximately 9.1 mM. Although this is higher than the protein concentration used experimentally, AFM images of pili on mica confirmed the MD-predicted ordering of FimA A80W A109W pili into one-layer bundles in the presence of 250 mM HMD (Fig. 3e).

To evaluate the conductivity of pili nanostructures at the micrometer scale, we first placed unordered networks of the FimA A80W A109W mutant, normalized for protein concentration, on interdigitated electrodes with 5μm non-conductive gaps between each pair of electrodes. We measured the conductance (Supplementary Fig. 5) and calculated the conductivity as described in methods. These pili networks show 100-fold higher conductivity than just HMD in the absence of pili at the micrometer scale, demonstrating remarkable long-range electron transfer (Fig. 3f). We then measured the conductivity of FimA A80W A109W nanostructures and found that assembling the pili networks into bundled nanostructures further increased conductivity by 5-fold,

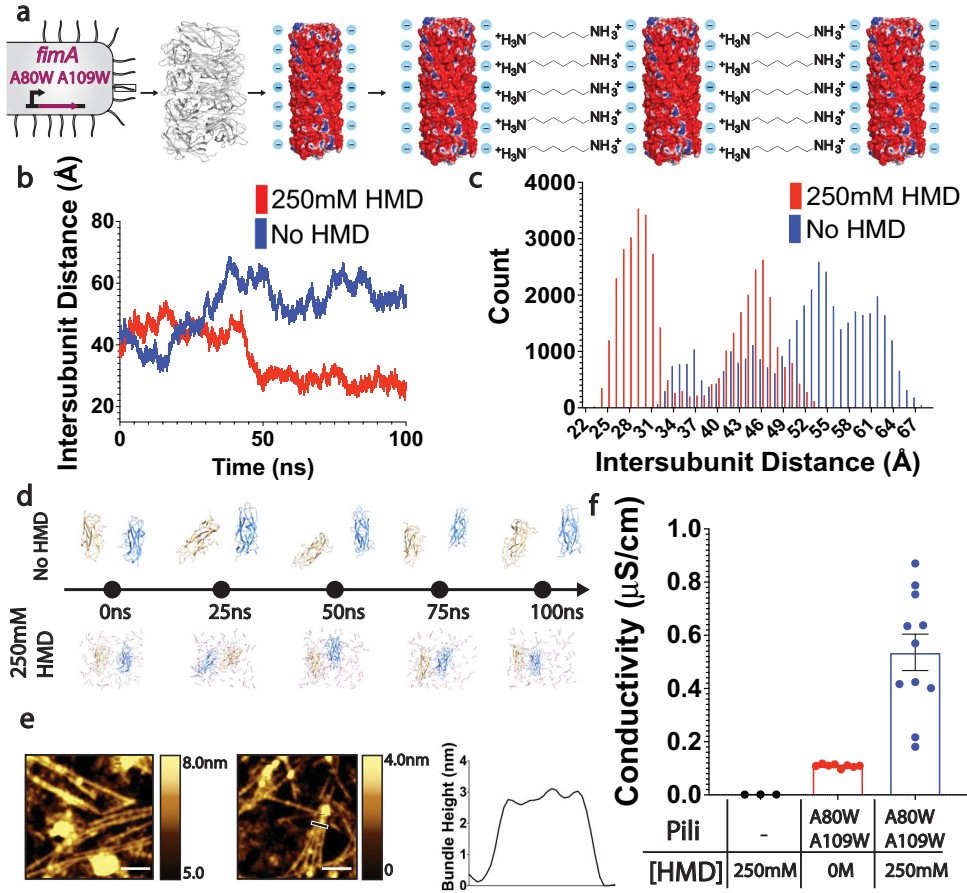

**Fig. 3 Computationally-guided design of hierarchical nanostructures show enhanced conductivity over micrometers. a** Strategy to align pili using HMD molecule **b** Time evolution of the distance between the geometric centers of each monomer. **c** Histogram displaying the distribution of distances between the geometric centers of the pilin monomers in the presence and absence of 250 mM HMD. Data was collected from separate 100 ns simulations. **d** Time evolution of the interaction between two FimA monomers in presence and absence of 250 mM HMD. **e** AFM images of pili on mica. Scale bar, 200 nm. Height profile (right) of pili at location shown (black bar crossing bundle) in middle image confirms the bundling. **f** Conductivity comparison of ordered pili. Error bars represent s.e.m. ($n = 3$).

from $0.109 \pm 0.0023$ μS/cm to $0.535 \pm 0.06$ μS/cm (Fig. 3f). The measured conductivity of bundled FimA A80W A109W nanostructures is comparable to previous measurements of conductive microbial nanowire networks[33]. It is important to note that the conductivity of individual filaments and the conductivity of bulk films is not comparable, and that film conductivity is always lower than the conductivity of individual pili due to high contact resistance either between pili or between pili and electrodes. Our results are consistent with previous studies on Geobacter protein nanowires that show film conductivity of 6 μS/cm[33] while individual filament conductivity is 0.5 mS/cm[11,36]. Notably, other networks of conductive proteins comprised of engineered curli fibers have shown a significantly lower film conductivity than our pili[19,40]. Although other self-assembly structures have been constructed by using different proteins, their conductivity has not been measured[41].

**Sequence-controlled synthesis of hybrid organic–inorganic pili increase conductivity ~170-fold.** We next sought to determine if the generation of a hybrid organic-inorganic pili biomaterial through the site-specific incorporation of nsAAs capable of click chemistry conjugated to AuNPs could further increase conductivity of type 1 pili. Using the GRO and the previously characterized pAzFRS.2.t1 orthogonal translation system (OTS)[23,25], we tested the incorporation of select nsAAs into pili (Supplementary Table 2). Previous work reported the nsAA

incorporation efficiency of pAzFRS.2.t1 used in this work to be >95%[25]. We chose nsAA molecules with high aromaticity (3-(2-Naphthyl)-L-alanine: [2NaA]) or those compatible with Cu-catalyzed click chemistry (4-azido-L-phenylalanine [pAzF], PrOF). Guided by the modified pili containing the aromatic amino acids, we tested for mutant pili expression using a yeast agglutination assay[24,30] (Supplementary Fig. 6) and found that 2NaA and pAzF could be incorporated at FimA position 80 but not position 109, whereas PrOF (Fig. 1d) could be incorporated at FimA position 109 but not 80. Incorporation of one 2NaA residue per FimA monomer increased pili conductivity ~5-fold to $2.71 \pm 0.16$ mS/cm (Fig. 4c, e), which although a substantial increase, is significantly less than the conductivity of all other mutant type 1 pili incorporating natural aromatic amino acid mutations (Figs. 2e, 4c, e).

We chose pili with PrOF inserted at position 109 in every FimA monomer as a scaffold for site-specific conjugation of AuNP to construct a hybrid organic-inorganic biomaterial. AuNPs, whose surface was covered with terminal azide groups, were conjugated to the terminal alkyne moieties of the PrOF residues (Fig. 1d) through the highly efficient Cu-catalyzed click chemistry reaction[42,43] (Supplementary Fig. 7). Using AFM imaging we found that PrOF-pili reacted with azide-AuNPs in the presence of copper were decorated with AuNPs (Fig. 4a, Supplementary Fig. 7) whereas PrOF-pili reacted with azide-AuNPs without the copper catalyst had no AuNPs attached

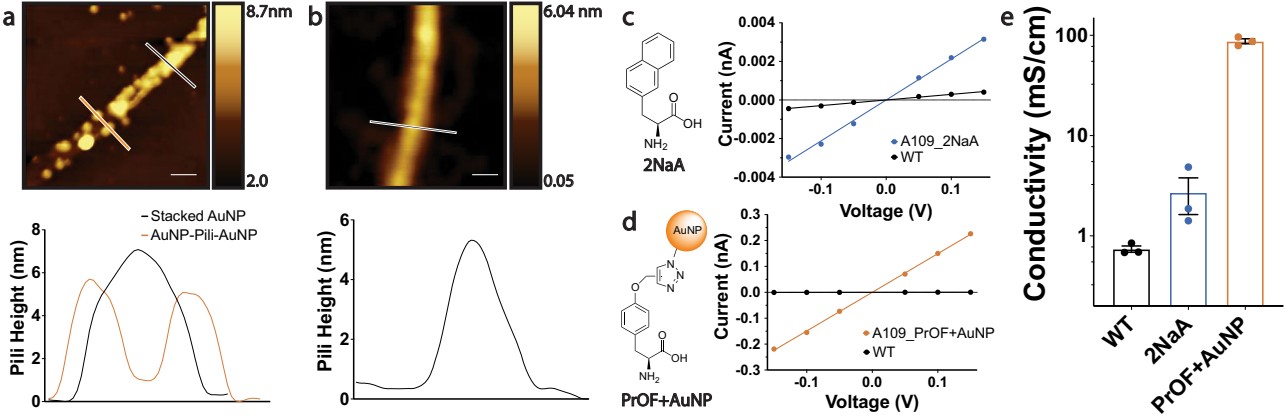

**Fig. 4 Hybrid organic-inorganic nanowires with ~170-fold higher conductivity through site-specific incorporation of nsAAs conjugated to gold nanoparticles (AuNPs). a** AFM image of AuNP-pili resulting from reacting azide-functionalized AuNPs with PrOF-containing pili with copper added to the Cu-catalyzed click chemistry reaction. Scale bar 100 nm, corresponding height profile below. Experiment independently repeated three times with similar results simultaneously with experiments performed for Fig. 4b. **b** AFM image of naked PrOF-containing pili resulting from reacting azide-functionalized AuNPs with PrOF-containing pili without copper added to the Cu-catalyzed click chemistry reaction. Scale bar 20 nm, corresponding height profile below. Experiment independently repeated three times with similar results simultaneously with experiments performed for Fig. 4a. **c, d** Current-voltage profile of pili incorporating **c** 2NaA and **d** PrOF conjugated with AuNP. Representative points were shifted by a constant value such that the slope of the current-voltage curve retained the same value but intercepted at zero for comparison purposes. Raw current-voltage data is provided in Supplementary Fig. 3 and the source file. **e** Conductivity of pili biomaterials incorporating nsAAs. Error bars represent s.e.m ($n = 3$).

(Fig. 4b). These results show that this conjugation proceeds through the selective Cu-catalyzed click reaction between the incorporated nsAA PrOF residues in the pili and the azide moieties on the AuNPs. The AFM images of the AuNPs conjugated along the pili filament (Fig. 4a, Supplementary Fig. 7) provide direct visual evidence of efficient and precise conjugation of AuNPs at PrOF residues. Interestingly, the height of pili reacted with copper decreased from 6 nm to 2 nm, potentially due to physical stress from reacting with AuNPs or copper ions altering pili structure (Fig. 4b, Supplementary Fig. 7c).

Finally, we placed the engineered pili conjugated to AuNPs on gold electrodes separated by 300 nm nonconductive gaps and isolated individual filaments bridging two electrodes using AFM to measure conductivity of the engineered individual filaments as described above. The AuNP-pili hybrid filaments were found to have conductivity higher than that of all other measured filaments at 87.40 ± 8.91 mS/cm (Fig. 4d, e), an increase of ~170-fold (Fig. 4d, e). Taken together, the high accuracy of encoding nsAAs[23,25], the high efficiency of the Cu-catalyzed click reaction[42,43], and the AFM images demonstrating the conjugation of AuNP to the PrOF residues demonstrate the construction of hybrid organic-inorganic biomaterials with significantly enhanced conductivity.

## Discussion

Our work combines the precise, site-specific engineering of bacterial nanowires enabled using synthetic biology methods with highly accurate conductivity measurements on individual filaments and filament nanostructures to demonstrate the production of multi-functional, highly conductive pili biomaterials. Guided by cryo-EM structures[9,17], we used MAGE to generate a targeted combinatorial library of genomic type 1 pili mutants to screen for aromatic amino acid mutations that retain pili self-assembly and exhibit increased conductivity. Notably, pili mutants exhibited a wide range of electronic conductivities based on the insertion site and aromaticity of the amino acids inserted into the pili. The insertion of aromatic amino acids into type 1 pili increased pili up to 84-fold with the double tryptophan mutant. Our use of single-filament conductivity measurements allowed us to directly ascribe an increase in conductivity of a filament to the incorporation of aromatic amino acids, providing

a higher level of engineered pili characterization compared to commonly used film measurements. The use of molecular dynamics simulations demonstrated more frequent association between pilus proteins in the presence of HMD, which informed the construction of hierarchical multidimensional nanostructures that demonstrated the efficient transport of charges over micrometer distances under ordinary thermal conditions. Bundling of pili increased conductivity 5-fold over the micrometer scale, implying that bundling may further facilitate electron transport down the filaments.

Finally, using the genomically recoded organism (GRO) and a highly efficient orthogonal translation system (OTS) able to incorporate nsAAs into proteins with >95% efficiency[25], we inserted 2NaA, pAzF, and PrOF at genetically encoded locations within the FimA monomer and thus across the entire surface of the pilus. We then conjugated azide-covered AuNPs to the terminal alkyne groups on PrOF residues in pili using the Cu-catalyzed cycloaddition "click" chemistry reaction. Using PrOF to directly label pili proteins with AuNP at single-residue precision is a strategy presented herein. The use of GROs and an efficient OTS to incorporate "click"-functional nsAAs into a bacterial nanowire allowed us to produce a class of hybrid organic-inorganic biomaterials which would otherwise be difficult to produce at scale without recoded bacteria capable of efficiently incorporating nsAAs. The efficient and site-specific incorporation of PrOF into individual FimA monomers allowed pili, large filaments made of polymerized FimA protein, to be uniformly covered in AuNPs—this advance demonstrates the ability to site-specifically functionalize large, micrometer-scale protein structures by re-purposing open codons (e.g., UAG) to encode nsAAs at high efficiency in a recoded bacterial host. This functionalization allowed the pili to be used as a chassis for a hybrid organic-inorganic material engineered with conductivity ~170-fold higher than wild-type.

Looking forward, our studies demonstrate the potential to use recoded *E. coli* strains for the production of genetically encoded hybrid biomaterials comprising organic and inorganic components. We have demonstrated the capability to endow functionality into pili using multisite incorporation of nsAAs with diverse chemical modalities. Building on our work, a large and diverse

array of chemical groups can be "clicked" onto pili-based materials. For example, taking inspiration from the stacked hemes in conductive *G. sulfurreducens* filaments[11,12], hemes modified with click-able moieties can be uniformly attached down the length of the pili chassis as another method of increasing of filament conductivity. As our approach to generate hierarchical structures can lead to multidimensional assembly of pili, future studies can incorporate photoreactive, cross-linkable nsAAs such as pAzF[44] into pili to generate light-activated arrays of conductive protein biomaterials. Furthermore, since our engineered pili are almost identical to those used previously to create pili nanostructures[22], other programmable assemblies, with varying effects on conductivity, are possible using our conductive pili. Applied work could examine the use of these electronically active pili in device applications such as sensors, transistors, and capacitors that span biological and electrical systems. More broadly, the capability of using GROs to produce multifunctional, structurally complex materials can be expanded toward the development of a programmable class of genetically encoded biomaterials with diverse chemistries and functions.

## Methods

**Bacterial strains**. All protein expression was done in strains derived from *E. coli* C321.A fimBE::tolC (strain A), where the fimBE genes were deleted using λ-red recombination by replacing them with a tolC cassette. *E. coli* C321.A fimA::gent (strain B) was created by deleting fimA with gentamycin using λ-red recombination. fimA mutants in the fimBE::tolC background were created using multiplex automated genome engineering (MAGE)[31]. *E. coli* strain NEB5α was used for cloning and plasmid assembly (New England Biolabs).

**Growth conditions**. When necessary, chloramphenicol, kanamycin, and gentamycin were used at 30, 20, and 5 μg/mL, respectively. SDS was used at 0.005% w/v for fimBE::tolC selection. All C321 cells were grown in Luria-Bertani broth containing 5 g/L NaCl supplemented with relevant antibiotics or SDS. NEB5α cells were recovered in SOC medium as described in the NEB5α competent cell protocol (New England Biolabs).

**Plasmids used**. To create plasmid pSHDS.1, we cloned out via PCR the fimA-fimH pili operon from the chromosome of genetically recoded organism *E. coli* C321.A (C321) and then used Gibson assembly to insert fimA-H into a plasmid based on pZE21G[23] under control of an aTc-inducible promoter. Plasmids pSHDS.80 and pSHDS.109 were made by creating mutations fimA A80TAG or A109TAG in pSHDS.1, respectively, using the Q5 site-directed mutagenesis protocol (New England Biolabs). Plasmid pAzFRS.2.t1[25] (addgene: https://www.addgene.org/73546/) was used to express the orthogonal translation system (OTS) that allows for the incorporation of nonstandard amino acids (NSAAs) into proteins.

**Multiplex Automated Genome Engineering (MAGE) and λ-RED recombination**. MAGE and λ-RED recombination were conducted as described elsewhere[31]. In short, liquid cultures were inoculated from frozen stock and grown overnight. These cultures were back-diluted 1:100 and grown to mid-logarithmic growth (OD$_{600}$ ~0.6) in a shaking incubator at 34 °C. λ-red recombination proteins Exo, Beta, and Gam were expressed by keeping the cells shaking in a water bath at 42 °C for 15 minutes. Cells were immediately chilled on ice and moved to a 4 °C environment. 1 mL of cells was centrifuged at 16000 × g for 15 s. The supernatant was removed, and the cells were resuspended in milli-Q water. The cells were again spun down, the supernatant was removed, and the cells resuspended in fresh milli-Q water to wash. This process was repeated three times. After the final spin, the supernatant was removed, and either mutagenic MAGE oligos prepared at 5-6μM in DNase-free water or 50 ng of dsDNA were added directly to the cell pellet and mixed thoroughly. The oligo-cell mixture was applied to a pre-chilled 1 mm gap electroporation cuvette (Bio-Rad) and electroporated at 1.8 kV, 200 V and 25 mF. The cells were immediately resuspended in 2 mL LB broth and recovered at 34 °C in a shaking incubator for 4 hours. When the cells again reached mid-logarithmic growth additional MAGE cycles were conducted or the cells were plated for future analysis.

Successful incorporation of mutations in the fimA gene of C321 fimBE::tolC was screened using MASC-PCR as described elsewhere[31].

dsDNA designed to replace fimBE with tolC and fimA with gentR contained the tolC or gentR cassettes with overhangs to regions directly before and after fimBE or fimA. Successful incorporation of tolC or gentR was confirmed by selection on SDS or gentamycin and sequencing of the fimBE and fimA regions.

**Oligonucleotides**. Oligonucleotides, including primers and MAGE oligos, were purchased from Keck Oligonucleotide Laboratory at Yale University.

**Chromosomal pili expression**. *WT* pili or those containing standard amino acid mutations (F, Y, or W) encoded genomically in strain A were expressed by inoculating three 500-mL flasks of LB from frozen glycerol stocks and growing the cultures at 34 °C without shaking. These cultures were grown in such conditions for 48 hours, after which cells were either imaged with TEM to visualize pili production, assayed for pili production using yeast agglutination, or used for pili protein harvesting and subsequent purification.

**Nonstandard amino acid incorporation into pili and subsequent expression**. Pili with inserted 4-propargyloxy-L-phenylalanine (PrOF) as a fimA A109PrOF mutation were expressed as follows. Strain B was transformed with both plasmid pSHDS.109 and pAzFRS.2.t1 and inoculated into LB containing chloramphenicol and kanamycin and grown overnight at 34 °C shaking at 225 rpm. The confluent culture was added as a 1:20 dilution to LB supplemented with chloramphenicol and kanamycin. To this culture, 20% w/v arabinose was added to a final concentration of 0.2% arabinose to induce the OTS encoded by pAzFRS.2.t1, 100 mM PrOF in 0.2 M NaOH was added to a final concentration of 1 mM PrOF, and 1 N HCl was added to a final concentration of 2 mM. The culture was grown at 34 °C shaking at 225 rpm for 3 h to induce the OTS encoded by pAzFRS.2.t1, after which anhydrotetracycline (aTc) was added to a final concentration of 60 ng/μL to induce pili containing fimA A109PrOF from pSHDS.109. Then, the culture was grown at 34 °C shaking at 135 rpm for an additional 8 h, after which cells were either imaged with TEM to visualize pili production, assayed for pili production using yeast agglutination, or used for pili protein harvesting and subsequent purification.

**Yeast agglutination assay for pili production**. 100 μL of yeast grown overnight from frozen stock and diluted to an OD$_{600}$ of ~1 in PBS pH 7.4 (−) MgCl$_2$ (−) CaCl$_2$ was mixed with 100 μL of *E. coli* grown in pili-producing conditions diluted to an OD$_{600}$ of ~1 in PBS pH 7.4 (−) MgCl$_2$ (−) CaCl$_2$. The mixture was incubated at room temperature and shaken at 350 rpm for 15 min. 1.5 μL of mixture was pipetted onto a glass slide and the mixture imaged using a standard optical microscope. Visible agglutination of yeast cells indicated production of pili by *E. coli* cells. Agglutination was detected qualitatively, and pili production was confirmed with TEM imaging of *E. coli* samples which produced an agglutination phenotype. Adapted from Firon et al[45].

**Pili purification**. Cells grown in pili producing conditions were spun down in 4 °C at 6000 rpm for 10 min and resuspended in 150 mM ethanolamine (ETA) pH 10.5. Cells were vortexed in a 50 mL conical tube for 2 min at vortex level 9 to shear pili from the cells, after which the cells were centrifuged in 4 °C at 10,000 rpm for 45 minutes. The supernatant was kept, and the cell pellet was discarded. The supernatant was then centrifuged in 4 °C at 23,000 × g for 1 h to remove remaining contaminants. The supernatant of this spin was kept, and the debris pellet was discarded. Saturated ammonium sulfate (SAS) (purchased from Thermo Fisher) was added to the supernatant to 15% of the final volume to precipitate out pili proteins. The solution was left static at 4 °C overnight, and then spun in 4 °C at 23,000× g for 1 h to collect precipitated pili. The supernatant was discarded, and the pili pellet was resuspended in ~600 μL 150 mM ETA pH 10.5. To this sample, 1 M MgCl$_2$ was added to a final concentration of 100 mM MgCl$_2$ and incubated at 4 °C overnight to precipitate out pili. The solution was then spun in 4 °C at 17,000 × g for 1 h to pellet pili. The supernatant was discarded, and the pili pellet was resuspended and solubilized in 150 mM ETA pH 10.5 by gentle pipetting to a volume of 150–500 μL depending on pellet size. The pili sample was then imaged or used for conductivity measurements. To determine purity of the sample via SDS-PAGE, the following steps were followed. To the pili sample solubilized in 150 mM ETA pH 10.5 in the previous step crystalline urea was added until the concentration was 6 M and the solution was kept at room temperature for 4 hours. The solution was then eluted through a 20 mL Sepharose column using 6 M urea as the eluent. The first 5 mL was taken as the void volume, followed by 15 fractions, each being 1 mL in volume. Fraction 2 was found to contain the purified pili filaments. Fraction 2 was then concentrated to 60 μL using a 3kDa-cutoff Amicon Ultra-0.5 mL Centrifugal Filter. 15 μL of the concentrated fraction 2 sample was used for further gel analysis as follows. To the 15 μL sample 5μL of 4x Laemmli buffer with 5% Beta-mercaptoethanol (BioRad) was added. The sample was then boiled for 25 minutes and then run on a SYPRO Ruby-stained SDS-PAGE gel. Once the presence and purity of FimA was confirmed, the remaining 45 μL volume of concentrated fraction 2 was then increased to 250 μL with 6 M Urea and subsequently dialyzed into 150 mM ETA pH 10.5. This protocol is partially inspired by a similar purification methodology[46].

**Transmission electron microscopy (TEM) imaging of pili samples**. Carbon film copper grids with mesh size 400 (Electron Microscopy Sciences) were cleaned with a PlasmaFlow plasma cleaner on medium for 30 s. 5 μL of pili sample was then dropcast onto the copper grid and left to adhere to the grid for 10 min, after which the remaining buffer was blotted off using filter paper. The samples were stained with a 1% PTA stain pH 6 by floating the grids on 50 μL droplets of stain for 30 s,

removing the grid and blotting the stain off with filter paper, floating the grids on the stain a second time for 30 s, and finally removing the grids and blotting the stain off with filter paper. The grids were then air dried for 10 min before storage and imaging.

**Mica sample preparation**. A small square of mica (Electron Microscopy Sciences inc.) was attached to tape and the top layer was peeled off, leaving an atomically flat fresh layer of mica. Onto this layer of mica, 3–4 μL of pili solution was dropcast and left to dry in a desiccator at 20% humidity overnight. The buffer was removed by washing the electrode surface with 17 μL Milli-Q water. Washing was done by dropping 17 μL of water onto the mica surface, gently pipetting the bubble of water up and down a few times, pipetting off the water, and discarding the dirty water and tip. This process was repeated three times. After the third wash, the mica sample was left to air dry for at least 45 min, after which pili were imaged using an Asylum Cypher ES atomic force microscope (Asylum Research).

**Fluorescence measurements**. A time-based assay was conducted on a Cary 3E UV-Vis Spectrophotometer with excitation wavelengths at 280 nm and 295 nm. The excitation bandwidth was 2.5 nm, with the emission bandwidth at 10 nm. The emission scan range for the 280 nm excitation was 295 nm to 500 nm and the emission scan range for the 295 nm excitation was 310 nm to 500 nm. The step size for the scan was 1 nm. The scan rate was 100 nm/min. The sample used was 125 μL of FimA A80W A109W pili and 125 μL FimA WT pili, dissolved in milli-Q water (pH 7.0) or 150 mM ETA pH 10.5. The emission intensity at each wavelength in the scan range was measured in counts per second. The background emission of the solvent at 125 μL was collected and subtracted to yield the final results.

**Electrode device sample preparation**. Electrode devices were washed with acetone, isopropanol, ethanol, and Milli-Q water, in that order, two times. After the second water wash, the device was left to air dry. The device was then plasma cleaned with the electrode side facing up using a tabletop Harrick Plasma cleaner on low for 2 min. The electrode was taken out of the plasma cleaner, and 3 μL of pili sample was immediately dropcast onto the center of the electrode, which was then left to dry in a desiccator at 20% humidity overnight. The buffer was removed by washing the electrode surface with 17 μL Milli-Q water three times. After the third wash, the electrode was left to air dry for at least 45 min. Electrodes bridged by *E. coli* pili were located using (Asylum Research).

**Atomic force microscopy (AFM) imaging**. Soft cantilevers (AC240TS-R3, Asylum Research) with a nominal force constant of 2 N/m and resonance frequencies of 70 kHz were used. The free-air amplitude of the tip was calibrated with the Asylum Research software, and the spring constant was captured by the thermal vibration method. The sample was imaged with a Cypher ES scanner using intermittent tapping (AC-air topography) mode. Images were analyzed using Asylum research v.16 and Gwyddion v.2.55.

**Pili conductance measurements for individual filaments**. Conductance measurements were performed as described previously[11]. Electrode devices with 17 electrodes spaced 300 nm apart were imaged under AFM to find individual pili filaments bridging two electrodes. Devices with pili bridging two electrodes were re-hydrated by dropping 0.3 μL of 150 mM ETA pH 10.5 onto the electrode and waiting 45 minutes. Conductance $G$ of pili was measured in a 2-electrode configuration inside a shielded dark box using an MPI Corporation probe station connected to a semiconductor parameter analyser (Keithley 4200A-SCS). DC voltages from −0.15 to 0.15 V in increments of 0.05 V were applied between electrodes bridged by pili and current $A$ was measured until a steady state was observed, typically over a period of two minutes. The linearity of the I-V characteristics was maintained by applying an appropriate low voltage and the slope of the I–V curve was used to determine the conductance (G). IV linear fits did not always go through (0,0), however conductivity was derived from the slope of the IV linear fit. Measurements were performed at low voltages (<0.15 V) and over longer times (>120 s) to ensure a lack of electrochemical leakage currents or faradic currents as evidenced by the absence of significant DC conductivity in buffer. All analysis was performed using IGOR Pro v.7 (WaveMetrics Inc.). Data was graphed using Graphpad Prism v.8. To display representative current-voltage curves in Figs. 2 and 4, the y-intercept of the raw linear fit was subtracted from the value of each data point such that all linear fits for the data had a y-intercept of zero. This display method preserves the slope of the linear fit and thus the conductance measurement of the filament, and allows for easy comparison of the differences in the magnitude of the conductance between filaments. Raw current-voltage data for all filaments is reported in Supplementary Fig. 3 and the source data file.

**Pili conductivity calculations for individual filaments**. The conductivity σ of filaments was calculated using the relation described elsewhere[33] $\sigma = \frac{GL}{a}$ where $G$ is conductance, $L$ is length of the filament measured between the electrodes, $a = \pi r^2$ is area of cross section of the filament with $2r$ as the height of the filament as measured by AFM, and $n$ is number of filaments bridging the electrodes in series. For pili with gold nanoparticles attached, the height was measured as pili + AuNP,

as in each case the gold nanoparticles completely covered the part of the filament between electrodes. All analysis was performed using IGOR Pro v.7 (WaveMetrics Inc.) Data was graphed using Graphpad Prism v.8.

**Pili bundling by hexamethylenediamine**. Three different concentrations of hexamethylenediamine, 0 M, 0.08 M, and 0.25 M, were mixed with pure samples of FimA A80W A109W pili in 150 mM ETA at pH 10.5. At this pH, both amines of the hexamethylenediamine molecule are still protonated[47]. These values were chosen based on the concentrations of hexamethylenediamine used elsewhere[22]. One original sample of FimA A80W A109W pili was divided among the three hexamethylene concentrations to normalize pili concentration between samples. These solutions were stored at 4 °C for 7 days in order to allow bundles to form. Afterwards, the solutions were imaged using AFM to determine if bundles were present. 0 M hexamethylenediamine was used as the negative control for no bundling, 0.08 M was found to be too small for bundling in WT pili, and 0.25 M was found to cause bundling in WT pili.

**Pili conductance measurements for bundles**. An interdigitated electrode (IDE) with a 5 μm spacing between the finger electrodes was cleaned with acetone and dried using nitrogen gas. 0.5 μL of the FimA A80W A109W pili sample was drop cast onto the IDE and the sample was placed in a desiccator for 30 minutes to dry. Then, 5 μL of milli-Q water was added to the area where the pili were dropcast. Water was left on the sample for 1 minute, after which it was wicked away with filter paper. The sample was placed in the probe station, and the current was measured between −0.2 and 0.2 V in increments of 0.05 V over 150 s using the same setup as for measuring individual filaments. The conductance $G$ of the pili network was found using the same method as for individual filaments.

**Pili conductivity calculations for bundles**. Conductivity was calculated by using the thin-film measurement formula described elsewhere[33], $\sigma = \frac{GL}{A}$, where $G$ is the conductance of the film, $L$ is the length of the gap of the electrode, 5 microns, and $A$ is the size of the drop that covers the electrode, which, after measurement, was estimated to be 0.926 μm$^2$.

**Synthesis of gold nanoparticle-pili (AuNP-pili)**. The pili A109PrOF incorporating an NSAA, PrOF, containing an alkyne, were used to produce the AuNP-pili. 5 nm NHS-activated gold nanoparticles (Cytodiagnostics) were reacted with 11-Azido-3,6,9-trioxaundecan-1-amine (Sigma Aldrich) using the recommended protocol from Cytodiagnostics. In the last step, AuNP-linker-azide were obtained by buffer exchanging with PBS pH 7.4 (−) MgCl₂ (−) CaCl₂ using 100 kDa Amicon Ultra centrifugal filter units (three times, centrifuging at 14,000 x g for 15 min). Next, AuNP-linker-azide were coupled onto the pili A109PrOF through copper-catalyzed azide-alkyne cycloaddition reaction. Briefly, 20 μL of 50 mM THPTA and 10 μL of 20 mM CuSO₄ were premixed for 30 min at room temperature. Later, 10 μL of 150 μM pili were diluted in 30μL of PBS pH 7.4 (−) MgCl₂ (−) CaCl₂ and 2.5 μL of AuNP-linker-azide were added to the mixture. Then, 1 μL of the mixture of THPTA/CuSO4 was added to the entire 40μL pili/AuNP mixture, are combined. Finally, 2.5 μL of 100 mM aminoguanidine and 2.5 μL of 100 mM sodium ascorbate are added. The reaction mixture was shaken at 500 rpm tabletop shaker (Santa Cruz Biotechnology) for 1 h. To stop the reaction and obtain the AuNP-linker-pili, the samples were buffer exchanged to 150 mM ETA pH 10.5 using 100 kDa Amicon Ultra centrifugal filter units (three times, centrifuging at 14,000 × g for 15 min).

**MD Simulation of pili bundling**. The atomic structure of the FimA monomer was obtained from the protein data bank (PDB ID 6C53). Polymerization of FimA into the type I pilus involves the insertion of an N-terminal beta strand formed by the first 20 residues of the neighboring subunit into an open groove in the immunoglobulin-like fold formed by residues 20 to 158 of the subunit of interest[17]. To construct our simulation model, we truncated the first 18 residues and added residues 3 to 20 of the neighboring subunit resulting in a complete immunoglobulin-like fold without an extraneous beta strand. Two of these monomer subunits were separated by a distance of 40 Å and rotated such that their outer surfaces faced each other. The hexamethylenediamine (HMD) molecule was parameterized using the CHARMM-GUI[48] and the CHARMM[49] general force field in its fully protonated state with a charge of +2. For our simulation with 250 mM HMD, we added 55 HMD molecules around the protein, calculated as the number of HMD molecules required to solvate the box with 250 mM. Simulation systems both with and without HMD were solvated in a TIP3P water box with dimensions of 101 × 88 × 61 Å using the VMD[50] solvate plugin. This provides at least 12 Å between the edge of the box and the closest protein atom. The Particle-Mesh Ewald (PME) method[51] was utilized to calculate long-range electrostatic interactions with 90 grid points in the x-direction, 80 grid points in the y-direction, and 54 grid points in the z-direction with a 12 Å cut-off. This spherical cut-off also extends to Lennard-Jones parameters. The temperature was maintained using a Langevin thermostat, and the pressure was maintained by using a constant Nosé-Hoover method in which Langevin dynamics is used to control fluctuations in the barostat. The velocity Verlet algorithm was used with a time-step of 1 fs.

MD simulations were performed with NAMD v2.13[52] using the CHARMM36[53] force-field parameters with periodic boundary conditions. First, each system was minimized, followed by a 500 ps simulation of the water box and any HMD molecules with fixed protein atoms. The models were then equilibrated to 310 K in the NVT ensemble for 3.5 ns under harmonic restraints with a force constant of 0.1 kcal/mol to the amino acid sidechains and a force constant of 1.0 kcal/mol to the protein backbone. Production runs were then performed in an NPT ensemble for 100 ns with frames being written to the trajectory every 2.5 ps. Analysis was performed by computing the distance between the geometric centers of each monomer subunit using a custom Tcl script available as part of the supplementary information. All of the simulations were done on the Grace supercomputing cluster, a computational core facility located at Yale University's Centre for Research Computing.

**Reporting summary**. Further information on research design is available in the Nature Research Reporting Summary linked to this article.

## Data availability
Source data for all figures are provided with this manuscript as supplementary information and are available upon reasonable request from the corresponding authors. Source data are provided with this paper.

## Code availability
The Tcl script used to compute the distance between the geometric centers of the monomers is available with this manuscript as supplementary information and upon reasonable request from the corresponding authors.

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

## Acknowledgements

We thank members of the Isaacs and Malvankar labs for helpful feedback on this study. We also thank Sophia Yi for helping with conductance measurements and conductivity calculations. FJI gratefully acknowledges support from the National Institutes of Health (RO1GM1404810) and National Science Foundation (MCB-1714860). N.S.M. gratefully acknowledges support from the National Institutes of Health Director's New Innovator award (1DP2AI138259-01) and an NSF CAREER award (1749662).

## Author contributions

D.M.S., N.S.M. and F.J.I. conceived and designed overall study. P.J.D. and N.S.M. conceived and designed portion of study on computationally-guided hierarchical nanostructures. D.M.S. generated all pili variants, imaged pili proteins, collected electrical measurements, and analyzed data. G.M. executed and analyzed bundling experiments with help from C.S. .S.E.Y. performed AFM imaging and analyzed data. P.A-G. aided in construction of pili with nsAAs and functionalization by AuNPs. J.P.O'B and G.M. also performed pili purification and characterization. P.J.D. and G.M. carried out computational modeling and molecular dynamics studies. V.S.,Y.G. and A.S.M carried out all TEM imaging. K.V. assisted with incorporation of nsAAs into pili. D.V. designed and fabricated electrode devices. F.J.I. and N.S.M. supervised the project. D.M.S., N.S.M. and F.J.I. wrote the manuscript with input from all authors.

## Competing interests

D.M.S., S.E.Y., N.S.M. and F.J.I. are named inventors on a patent application filed by Yale University (U.S.S.N. 63/302,932) covering the methods of making and using engineered FimA proteins encoded with metal nanoparticles. F.J.I. is a co-founder at Pearl Bio. The remaining authors declare no competing interests.
