## [Peer Review File · Nature Communications]

Reviewers' Comments:

Reviewer #1:

Remarks to the Author:

The manuscript by Shapiro describes the engineering of bacterial pili to create electrically conductive nanowires. The authors substituted in aromatic residues into E. coli type 1 pili and demonstrated the mutant pili containing phenylalanine, tyrosine, or tryptophan had increased conductivity. This result agrees with previous published studies that have demonstrated enhanced electronic conductivity in protein assemblies upon aromatic residue incorporation.

It was interesting that the electron transfer was over a greater distance than previously described pi-pi stacking. The authors suggest electron hopping or tunnelling, however, I would have liked to see a deeper exploration in how charge transfer functions in these types of engineered materials. If the wild-type pili are not conductive, and pili are not involved in electron transfer in E. coli (as the authors argue), then what is the significance of engineering conductive pili?

Subsequently, the authors used a previously described approach to bundle the pili into larger structures. Addition of hexamethylenediamine was shown to cause pili-pili interactions through electrostatic interactions - presumably this is what the "programmable self-assembly" refers to in the manuscript title? I struggle to understand how this is actual programmability - can the authors tune the overall dimensions, e.g., length and widths, of the material? Is the assembly reversible? There are plenty of methods to assemble protein fibers, many of which can produce ordered nanostructures rather than these aggregates, so I have a hard time seeing the importance in this approach.

The authors then go on to incorporate non-standard amino acids. The incorporation of aromatic nsAA was interesting, and could perhaps lead to the creation of nanowires with greater conductivity than phenylalanine, tyrosine, or tryptophan-containing nanowires. Finally, the authors use nsAA to attach gold nanoparticles to the protein via a standard alkyne-azide "click" reaction to create gold-protein wires, which were much more conductive than the engineered aromatic nanowires

Reviewer #2:

Remarks to the Author:

I was reviewer 1 of the previous submission. I believe the authors carefully addressed my comments and I am in support of publication of the revised manuscript.

Reviewer #3:

Remarks to the Author:

The manuscript is significantly improved in the previous and all my original comments have been addressed. My comment to these that the authors MD simulations with HMD and FimA suggest that the FimA concentration would be approximately 10 mM, which is rather high for modelling protein interactions. I appreciate this is a limitation of MD, but is probably worth mentioning.

The other main comment is the discussion about conductive Pili. There is a group in the US who have also been studying the phenomenon of protein conductivity. It is worth discussing how this fits in with your current understanding of protein and pilli conductivity.

Zhang B, Song W, Brown J, Nemanich R, Lindsay S: Electronic Conductance Resonance in Non-Redox-Active Proteins. *J Am Chem Soc* 2020, 142:6432-6438

Below please find a point-by-point response to all Reviewer comments:

Point-by-point response (bulleted) to Reviewers' comments (*italicized*):

Reviewer #1 (Remarks to the Author):

The manuscript by Shapiro describes the engineering of bacterial pili to create electrically conductive nanowires. The authors substituted in aromatic residues into E. coli type 1 pili and demonstrated the mutant pili containing phenylalanine, tyrosine, or tryptophan had increased conductivity. This result agrees with previous published studies that have demonstrated enhanced electronic conductivity in protein assemblies upon aromatic residue incorporation.

1. *It was interesting that the electron transfer was over a greater distance than previously described pi-pi stacking. The authors suggest electron hopping or tunnelling, however, I would have liked to see a deeper exploration in how charge transfer functions in these types of engineered materials. If the wild-type pili are not conductive, and pili are not involved in electron transfer in E. coli (as the authors argue), then what is the significance of engineering conductive pili?*
 - Our work demonstrates a novel approach to engineer electronic functionality in proteins using both natural and nonstandard aromatic amino acids conjugated to gold nanoparticles. We have recently performed a detailed exploration of charge transfer in aromatic residues (Ref. 19). The focus on this work is to apply those principles, combined with synthetic chemistries enabled by synthetic biology and genetic code expansion technologies, to build novel organic-inorganic materials.

2. *Subsequently, the authors used a previously described approach to bundle the pili into larger structures. Addition of hexamethylenediamine was shown to cause pili-pili interactions through electrostatic interactions – presumably this is what the “programmable self-assembly” refers to in the manuscript title? I struggle to understand how this is actual programmability – can the authors tune the overall dimensions, e.g., length and widths, of the material? Is the assembly reversible? There are plenty of methods to assemble protein fibers, many of which can produce ordered nanostructures structures rather than these aggregates, so I have a hard time seeing the importance in this approach.*
 - These are excellent points and provide an opportunity for us to clarify unique aspects of this study. Our work is the first to demonstrate high electronic functionality in individual pili filaments as well their ordered assemblies whereas previous work was limited to ordering wild-type pili which lack conductivity (Ref. 20). Furthermore, we enhance conductivity by 150-fold than that found in wild-type pili (Fig. 2). Thus, we program electronic properties in these proteins using both natural and nonstandard amino acids conjugated to synthetic gold nanoparticles. We have further clarified this in the revised manuscript (page 6, lines 13-16, and page 9, lines 25-27).

The authors then go on to incorporate non-standard amino acids. The incorporation of aromatic nsAA was interesting, and could perhaps lead to the creation of nanowires with greater conductivity than phenylalanine, tyrosine, or tryptophan-containing nanowires. Finally, the authors use nsAA to attach gold nanoparticles to the protein via a standard alkyne-azide “click” reaction to create gold-protein wires, which were much more conductive than the engineered aromatic nanowires

Reviewer #2 (Remarks to the Author):

I was reviewer 1 of the previous submission. I believe the authors carefully addressed my comments and I am in support of publication of the revised manuscript.

- We thank the Reviewer for recommending publication of our revised manuscript.

Reviewer #3 (Remarks to the Author):

The manuscript is significantly improved in the previous and all my original comments have been addressed.

1. *My comment to these that the authors MD simulations with HMD and FimA suggest that the FimA concentration would be approximately 10 mM, which is rather high for modelling protein interactions. I appreciate this is a limitation of MD, but is probably worth mentioning.*
 - As suggested by the Reviewer, we have clarified this point in the revised manuscript (page 6, lines 22-25).
2. *The other main comment is the discussion about conductive Pili. There is a group in the US who have also been studying the phenomenon of protein conductivity. It is worth discussing how this fits in with your current understanding of protein and pilli conductivity. Zhang B, Song W, Brown J, Nemanich R, Lindsay S: Electronic Conductance Resonance in Non-Redox-Active Proteins. J Am Chem Soc 2020, 142:6432-6438*
 - As suggested by the reviewer, we have now included a description of other non-aromatic proteins that show conductivity on page 4, lines 9-12, as follows: “Indeed, several non-aromatic proteins have recently been shown to be conductive over nanometres in single-molecule measurements, with little decay due to distance, provided charge is injected into the protein interior with good contact. However, the conduction mechanism is unknown (J. Am. Chem. Soc. 2020, 142, 6432–6438).”